# Hypertriglyceridemia-associated acute pancreatitis: Response to continuous insulin infusion

**Vishnu Priya Pulipati**[1,2], **Ambika Amblee**[1,2]*, **Sara Elizabeth T. Yap**[2], **Hafeez Shaka**[2], **Bettina Tahsin**[1,2], **Leon Fogelfeld**[1,2]

1 Division of Endocrinology, John H. Stroger Jr. Hospital of Cook County, Chicago, Illinois, United States of America, 2 Divison of Endocrinology, Rush University Medical Center, Chicago, Illinois, United States of America

* aamblee@cookcountyhhs.org

**Data Availability Statement:** All relevant data are within the paper and its Supporting information files.

## Abstract

### Objective

To assess the response of serum triglycerides (TG) to continuous insulin infusion (CII) in adults with hypertriglyceridemia-associated acute pancreatitis (HTGP).

### Methods

Retrospective analysis of TG response to standardized CII therapy in 77 adults admitted to intensive care with TG >1000 mg/dL and HTGP.

### Results

Participants had initial TG 3869.0 [2713.5, 5443.5] mg/dL and were 39.3 ± 9.7 years old, 66.2% males, 58.4% Hispanic, BMI 30.2 [27.0, 34.8] kg/m², 74.0% with diabetes mellitus (DM) and 50.6% with excess alcohol use. TG-goal, defined as ≤1,000 ± 100 mg/dL, was achieved in 95%. Among the 73 TG-goal achievers (responders), 53.4% reached TG-goal in <36 hours after CII initiation (rapid responders). When compared to slow responders taking ≥36 hours, rapid responders had lower initial TG (2862.0 [1965.0, 4519.0] vs 4814.5 [3368.8, 6900.0] mg/dL), BMI (29.4 [25.9, 32.8] vs 31.9 [28.2, 38.3] kg/m²), DM prevalence (56.4 vs 94.1%), and reached TG-50% (half of respective initial TG) faster (12.0 [6.0, 17.0] vs 18.5 [13.0, 32.8] hours). Those with DM (n = 57) vs non-DM (n = 20) were obese (31.4 [28.0, 35.6] vs 27.8 [23.6, 30.3] kg/m²), took longer to reach TG-final (41.0 [25.0, 60.5] vs 14.5 [12.5, 25.5] hours) and used more daily insulin (1.7 [1.3, 2.1] vs 1.1 [0.5, 1.9] U/kg/day). Among those with DM, the rapid responders had higher daily use of insulin vs slow responders 1.9 [1.4, 2.3] vs 1.6 [1.1, 1.8] U/kg/day. All results significant. In multivariable analysis, predictors of faster TG response were absence of DM, lower BMI and initial TG.

### Conclusion

CII was effective in reaching TG-goal in 95% of patients with HTGP. Half achieved TG-goal within 36 hours. Presence of DM, higher BMI and initial TG slowed the time to reach TG-

**Funding:** The author(s) received no specific funding for this work.

**Competing interests:** The authors have declared that no competing interests exist.

goal. These baseline parameters and rate of decline to TG-50% may be real-time indicators to initiate and adjust the CII for quicker response.

## Introduction

Hypertriglyceridemia-associated acute pancreatitis (HTGP) is a serious and common disorder with no consensus on most effective management approach to mitigate the disease. Hypertriglyceridemia (HTG) is the third most common cause of acute pancreatitis (AP), accounting for approximately 10% of cases, after gallstones and alcohol [1]. The pathogenesis of HTGP is unclear but the most accepted mechanism suggests hydrolysis of TG by pancreatic lipase, in and around the pancreas, leading to the production of free fatty acids (FA) in pancreatic capillaries. These FAs accumulate in capillaries leading to pancreatic ischemia with acidosis, trypsinogen activation, and the initiation of AP [2].

The management of HTGP includes an aggressive reduction of TG to at least <1000 mg/dL using either continuous insulin infusion (CII) or plasmapheresis (PEX) based on clinical severity at presentation. The reduction of TG to less than 1000 mg/dL showed a decrease in the risk of further AP episodes [3]. Once the TG is at goal and the patient can tolerate oral intake, dietary modifications (diet low in refined carbohydrates and saturated and trans fat as well as no alcohol) and lipid-lowering medications (such as fibrates, omega-3 fatty acids and selected statins) are added to achieve long-term TG control [4].

CII is an effective modality of therapy to reduce TG levels rapidly. Insulin reduces TG levels by promoting the synthesis of lipoprotein lipase that hydrolyzes plasma TG and facilitates the storage of fatty acids in adipocytes [5]. In the currently available literature, there are only limited studies looking at the duration required for CII to reduce TG to <1000 mg/dL and no studies that measured the quantity of insulin required for reducing TG [6–8]. Our study aimed to fill these gaps.

## Methods

### Study design

This retrospective study examined adults admitted to the intensive care unit (ICU) of John H Stroger Jr Hospital of Cook County, in Chicago, Illinois from January 2011 to June 2019 with a TG >1000 mg/dL and HTGP requiring CII. Patients received a standardized 0.1 U/kg regular insulin bolus followed by the ICU adjustable insulin infusion protocol based on the serum glucose levels. The target glucose range was a 110–150 mg/dL with glucose levels checked hourly. The amount of insulin adjustment was depending upon the hourly infusion rates and the levels of glucose above or below the target range. Fluid management, using either 0.9% or 0.45% saline, was consistent for all ICU patients based on hydration parameters. We aimed to investigate the rate of the HTG response to CII in patients with HTGP and to identify the predictive factors that affected the response. The study, which analyzed de-identified data, was approved with a consent waiver by the Institutional Review Board of John H Stroger Jr Hospital of Cook County, Chicago, Illinois.

### Inclusion/Exclusion criteria

Inclusion criteria were adult patients (age ≥18 years) admitted to the ICU with the diagnosis of AP, admission serum TG level >1000 mg/dL, and who received treatment with CII. AP was

defined per 2012 Atlanta classification of acute pancreatitis as any two of the three following criteria: abdominal pain (acute onset, persistent, severe, epigastric pain often radiating to the back), pancreatic enzymes (serum lipase or amylase) level at least 3 times greater than the upper limit of normal, and/or computed tomography (CT) or magnetic resonance imaging (MRI) evidence of acute pancreatitis [9]. Exclusion criteria included pregnancy, patients who underwent therapeutic plasma exchange (TPE) procedure, and those who did not receive a bolus prior to CII. Data were obtained from a detailed electronic medical chart review.

## Outcomes

The primary outcome was the achievement of the TG goal ≤1,000 ± 100 mg/dL. Secondary outcomes were time durations to achieve TG-50% (half of the respective initial TG) and TG-goal.

## Variables

Data collected included baseline demographic and clinical information such as age, gender, race, body mass index (BMI), excess alcohol use, history of gallbladder disease, DM, hypothyroidism, and serum TG level on admission. Gallbladder sludge and gallstones were considered present if documented at admission or noted in any abdominal imaging (ultrasound or CT) during the hospital stay. Excess alcohol use was identified by self-report per Center for Disease Control (CDC) definition criteria [10]. DM was defined per the American Diabetes Association (ADA) criteria as HbA1C ≥6.5% or documented diagnosis of any type of DM [11]. Per World Health Organization (WHO) and CDC criteria, obesity was defined as BMI >30 kg/m$^2$ [12]. To evaluate the response of HTG to CII, other pertinent variables collected included documentation if the patient was placed on bowel rest (NPO or nothing per mouth) after admission, time of CII initiation, time of initiation of oral TG-lowering medications (such as fibrates, statins, fish oil) administered during hospitalization.

Serum TG levels while on CII used for analysis were three major reference points: 1) Initial TG, serum TG level at the start of CII, 2) TG-50%, serum TG level at the time of reaching half of respective initial serum TG, 3) TG-goal defined as reaching TG ≤1000 ± 100 mg/dL. For patients who did not reach the TG-goal, the lowest serum TG was defined as TG-final. Accordingly, two time durations were defined in hours: 1) time to reach TG-50% from initiation of CII and 2) time to reach TG-goal from initiation of CII. The CII was analyzed as 1) total dose of insulin used from the start of CII to TG-goal and 2) total daily insulin use per kilogram body weight (U/Kg/day). Frequency of TG checks of patients in the ICU was a minimum of every 6–12 hours until TG was at goal.

## Statistical analysis

Continuous variables were reported as mean ± standard deviation for parametric data and as median [25th, 75th percentile] for non-parametric data. Differences between continuous variables were measured by independent-samples t-test for parametric data and Mann-Whitney U tests for non-parametric data. Categorical variables were reported as n (sample size) and percentages. The association (or relationship) between categorical variables was measured by the chi-square ($\chi^2$) test. A 2-tailed P <0.05 was considered statistically significant. Binomial logistic regression was used to estimate confidence intervals, odds ratios (OR), and P values. The multivariate regression analysis was used to measure the degree of the linear relationship between independent variables (predictors) and dependent variables (responses). IBM SPSS, version 26 software (SPSS Inc, Chicago, Illinois) was used for all statistical analyses.

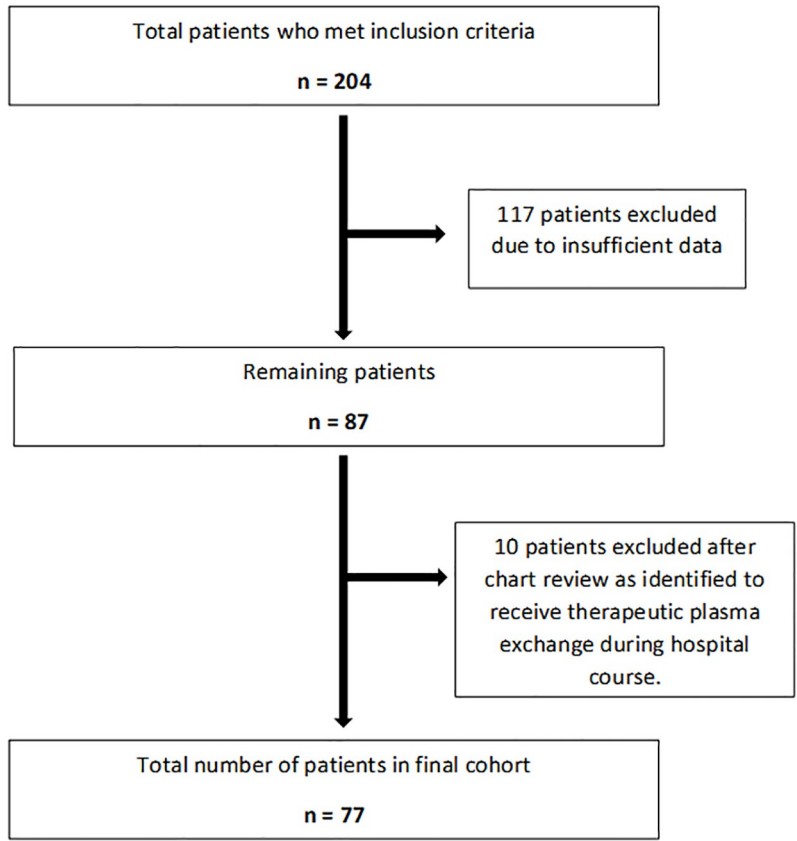

**Fig 1. Consort chart.**

## Results

Based on the inclusion criteria, a total of 204 patients were identified for the study (Fig 1). Out of those patients, 117 patients were excluded from the study due to insufficient medication administration record (MAR) data during the hospital stay. After chart review, another 10 patients were excluded as they received TPE during the hospital stay. A total of 77 patients were identified for the final analysis: 75 patients received CII only, the other 2 patients received CII and subsequent TPE due to poor response to CII. For the two patients included in the final analysis who received CII and TPE, only the data while on CII was included.

The baseline characteristics of the 77 patients who received CII are shown in Table 1. The initial TG was 3869.0 [2713.5, 5443.5] mg/dL. The average age was 39.3 ± 9.7 years, 66.2% were males, 58.4% Hispanic, BMI was 30.2 [27.0, 34.8] kg/m$^2$, 50.6% reported excess alcohol use, and 74% had DM. All patients were placed NPO after admission.

TG-goal was achieved in 73 patients (95%) (RP, responders). One patient with initial TG of 7035.0 mg/dL was classified as a responder with TG final of 1191.0 mg/dL since the CII was stopped at that level. The four patients who did not reach TG-goal (NR, non-responders) did not differ significantly at baseline from responders. Among the four NR, two underwent TPE and two were transitioned to oral fibrates as they clinically improved.

The time course to reach TG-goal in RP is shown in Fig 2. The time to reach the TG-50% and the TG-goal varied widely from 25th percentile to 75th percentile, 8.0 to 24.5 hours and

**Table 1. Baseline characteristics of the study population.**

| Baseline Characteristics | n = 77 |
|---|---|
| Age (years) | 39.3 ± 9.7 |
| Male, n (%) | 51 (66.2) |
| Ethnicity, n (%) | |
| Hispanic | 45 (58.4) |
| African American | 15 (19.5) |
| White | 15 (19.5) |
| Other | 2 (2.6) |
| BMI (kg/m$^2$) | 30.2 [27.0, 34.8] |
| DM, n (%) | 57 (74.0) |
| HbA1C% | 9.6 ± 3.4 |
| History of excess alcohol use, n (%) | 39 (50.6) |
| Gall stone disease, n (%) | 1 (1.3) |
| Overt hypothyroidism, n (%) | 2 (2.6) |
| Initial serum TG (mg/dL) | 3869.0 [2713.5, 5443.5] |

16.0 to 52.0 hours respectively. Among the 73 RP, patients were further divided per time taken to reach TG-goal as <36 hours (RRP, rapid responders, n = 39) vs ≥36 hours (SRP, slow responders, n = 34) as shown in Table 2. The SRP vs RRP were more obese (31.9 [28.2, 38.3] vs 29.4 [25.9, 32.8] kg/m$^2$, p = 0.028), more likely to have DM (94.1 vs 56.4%, p = 0.000), had higher initial serum TG (4814.5 [3368.8, 6900.0] vs 2862.0 [1965.0, 4519.0] mg/dL, p = 0.000), and took longer to reach TG-50% (18.5 [13.0, 32.8] vs 12.0 [6.0, 17.0] hours, p = 0.001).

Among the total 77 patients, 57 had DM and 20 did not have DM as shown in Table 3. Among the patients with DM, 35.1% were on subcutaneous insulin before hospitalization. Compared to non-DM, patients with DM were obese (31.4 [28.0, 35.6] vs 27.8 [23.6, 30.3] kg/m$^2$, p = 0.003), took longer time to reach TG-50% (16.0 [10.5, 27.0] vs 9.5 [5.3, 14.8] hours, p = 0.004) and TG-goal (41.0 [25.0, 60.5] vs 14.5 [12.5, 25.5] hours, p = 0.000) since starting CII and had higher daily insulin need (1.7 [1.3, 2.1] vs 1.1 [0.5, 1.9] U/kg/day, p = 0.041).

In bivariate analysis, the factors at baseline associated with faster TG response (TG-goal in <36 hours since starting CII) were the absence of DM, lower HbA1C, BMI, and initial TG. In binomial logistic regression analysis, only the absence of DM (OR 18.9, CI 3.02–119.2,

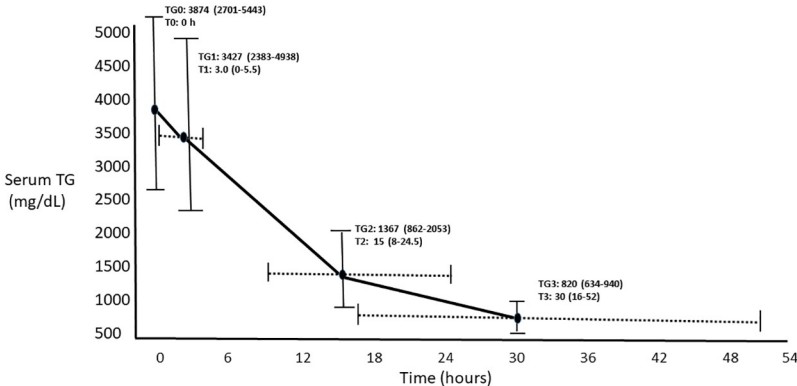

**Fig 2. Reduction in serum triglyceride levels over time after insulin initiation in the 73 responders.** TG0 and T0-Intial TG and time, TG1 and T1-TG and time as insulin initiation, TG2 and T2-TG and time for reduction by 50% or less, TG3 and T3-TG goal and time at insulin drip discontinuation.

**Table 2. Clinical and laboratory characteristics among the 73 responders who reached TG goal in less than 36 hours vs 36 hours or more.**

| | Responders (n = 73) | Less than 36 hours (n = 39) | 36 hours or more (n = 34) | P value[#] |
|---|---|---|---|---|
| **Demographics**: | | | | |
| Age (years) | 39.1 ± 9.7 | 39.3 ± 8.6 | 38.9 ± 11.0 | 0.881 |
| Males, n (%) | 49 (67.1) | 27 (69.2) | 22 (64.7) | 0.80 |
| Ethnicity, n (%) | | | | 0.36 |
| Hispanic | 45 (61.6) | 25 (64.1) | 20 (58.8) | |
| African American | 14 (19.2) | 6 (15.4) | 8 (23.5) | |
| White | 12 (16.4) | 8 (20.5) | 4 (11.8) | |
| Other | 2 (2.7) | 0 (0.0) | 2 (5.9) | |
| BMI (kg/m$^2$) | 30.2 [27.0, 34.8] | 29.4 [25.9, 32.8] | 31.9 [28.2, 38.3] | 0.028* |
| Excess alcohol use, n (%) | 37 (50.7) | 24 (61.5) | 13 (38.2) | 0.062 |
| DM, n (%) | 54 (74.0) | 22 (56.4) | 32 (94.1) | 0.000* |
| **Baseline**: | | | | |
| HbA1C% | 9.6 ± 3.5 | 8.4 ± 3.2 | 10.9 ± 3.3 | 0.002* |
| Initial serum TG (mg/dL) | 3874.0 [2701.0, 5443.5] | 2862.0 [1965.0, 4519.0] | 4814.5 [3368.8, 6900.0] | 0.000* |
| Serum TG at the start of CII (mg/dL) | 3427.0 [2383.0, 4938.5] | 2701.0 [1770.0, 3992.0] | 4122.0 [3248.5, 6543.0] | 0.000* |
| Time from initial TG to start of CII (hours) | 3.0 [0.0, 5.5] | 4.0 [0.0, 5.0] | 3.0 [0.0, 6.0] | 0.848 |
| **Midpoint**: | | | | |
| TG-50% (mg/dL) | 1367.0 [862.5, 2053.5] | 974.0 [692.0, 1367.0] | 1753.0 [1378.0, 2656.5] | 0.000* |
| Time from start of CII to TG-50% (hours) | 15.0 [8.0, 24.5] | 12.0 [6.0, 17.0] | 18.5 [13.0, 32.8] | 0.001* |
| Total insulin used to reach TG-50% (units) | 85.6 [38.3, 145.4] | 56.0 [29.3, 111.0] | 110.3 [78.5, 221.8] | 0.004* |
| **Final**: | | | | |
| TG final (mg/dL) | 820.0 [634.5, 940.0] | 752.0 [547.0, 889.0] | 898.0 [668.0, 966.3] | 0.013* |
| Time from start of CII to TG goal (hours) | 30.0 [16.0, 52.0] | 17.0 [12.0, 26.0] | 54.0 [43.5, 67.5] | 0.000* |
| Total insulin used to reach TG goal (units) | 171.0 [82.0, 266.9] | 87.0 [41.3, 175.0] | 266.9 [191.7, 469.3] | 0.000* |
| Total daily insulin used to reach TG goal (units/kg/day) | 1.6 [1.0, 2.1] | 1.8 [1.0, 2.3] | 1.5 [1.0, 1.8] | 0.169 |

*Statistically significant P <0.05.

[#]The p value pertinent to <36 hours vs ≥ 36 hours.

p = 0.002) and higher daily insulin dose per kg (OR 3.11, CI 1.19–8.15, p = 0.02) were related to faster TG response.

Among those with DM, the patients were divided into RRP and SRP (Table 4). In bivariate analysis and binary logistic regression, compared to SRP with DM, RRP with DM showed higher total daily insulin use (U/kg/day) (1.9 [1.4, 2.3] vs 1.6 [1.1, 1.8], OR 4.6, CI 1.4–15.3, p = 0.019).

After completion of CII therapy, 70.1% were converted to subcutaneous insulin. During the hospital stay, 92.2% received fibrates after 27.0 [9.0, 43.0] hours and 29.9% received statins 53.6 ± 36.6 hours after initiation of CII. Among the 73 RP, the use of fibrates <36 hours vs ≥36 hours from the start of CII did not show a significant difference (p = 0.325) in the time taken from initiation of CII to TG-goal. All patients were discharged from the hospital indicating that there was a resolution of acute pancreatitis.

## Discussion

Currently, there is no consensus concerning the most effective management of HTGP. Therapeutic options for acute HTGP management include supportive management for acute pancreatitis (including intravenous hydration, analgesics, bowel rest), CII, and, in select cases,

**Table 3. Differences between patients with and without diabetes.**

| | Patients with DM (n = 57) | Patients without DM (n = 20) | P value |
|---|---|---|---|
| **Demographics**: | | | |
| Age (years) | 38.6 ± 9.9 | 41.3 ± 9.0 | 0.279 |
| Males, n (%) | 38 (66.7) | 13 (65.0) | 0.892 |
| Ethnicity, n (%) | | | 0.028* |
| Hispanic | 35 (61.4) | 10 (50.0) | |
| African American | 14 (24.6) | 1 (5.0) | |
| White | 7 (12.3) | 8 (40.0) | |
| Other | 1 (1.8) | 1 (5.0) | |
| Excess alcohol use, n (%) | 26 (45.6) | 13 (65.0) | 0.136 |
| BMI (kg/m$^2$) | 31.4 [28.0, 35.6] | 27.8 [23.6, 30.3] | 0.003* |
| **Baseline**: | | | |
| Initial serum TG (mg/dL) | 3987.0 [2905.5, 6246.5] | 3426.5 [2186.5, 4519.0] | 0.099 |
| Serum TG at the start of CII (mg/dL) | 3427.0 [2520.0, 5539.0] | 2837.5 [2171.0, 4519.0] | 0.205 |
| Time from initial TG to start of CII (hours) | 3.0 [0.0, 5.0] | 4.0 [1.0, 6.7] | 0.136 |
| **Midpoint**: | | | |
| TG-50% (mg/dL) | 1470.0 [977.0, 2260.5] | 1157.0 [791.0, 1537.8] | 0.038* |
| Time from start of CII to TG-50% (hours) | 16.0 [10.5, 27.0] | 9.5 [5.3, 14.8] | 0.004* |
| Total insulin used to reach TG-50% (units) | 101.9 [63.5, 191.5] | 38.3 [18.6, 54.8] | 0.000* |
| **Final**: | | | |
| TG final (mg/dL) | 847.0 [641.5, 962.5] | 791.0 [586.0, 970.8] | 0.440 |
| Time from start of CII to TG goal (hours) | 41.0 [25.0, 60.5] | 14.5 [12.5, 25.5] | 0.000* |
| Required ≥36 hours to reach TG goal, n (%) | 35 (61.4) | 2 (10.0) | 0.000* |
| Total insulin used to reach TG goal (units) | 249.4 [129.9, 408.7] | 60.5 [30.5, 106.4] | 0.000* |
| Total daily insulin used to reach TG goal (units/kg/day) | 1.7 [1.3, 2.1] | 1.1 [0.5, 1.9] | 0.041* |

*Statistically significant P <0.05.

PEX [13]. Only a few studies compared CII against supportive management. In one study, there was no difference between the two groups and subcutaneous insulin was used in some participants of the conservative treatment group [6]. In Berberich et al., a subgroup analysis of conservative therapy only (n = 10) vs. those who concurrently received CII (n = 12), there was no difference in the rate of TG decline with the limitation of this conclusion being the small sample size [14].

PEX is an effective but cumbersome and expensive process that removes TG from circulation, reserved for select patients with severe life-threatening HTGP. After one session, PEX can decrease TG by approximately 70% [15]. Potential complications of PEX include urticaria, risk of infections, allergic reaction to donor plasma, hypotension, and central catheter-associated complications such as hemorrhage, thrombosis, and injury to blood vessels [13].

There are few head-to-head studies comparing the efficacy of CII and PEX in acute HTGP management [16, 17].

Growing evidence consistently demonstrates CII to be an effective, safe, economic, and minimally invasive therapeutic option in patients with HTGP with or without DM [7, 16–19]. CII is effective and non-inferior to PEX in achieving a rapid reduction in TG levels [16, 17] and is more feasible and easier to use than PEX.

Our retrospective study is unique in evaluating the predictors as well as the rate of response of HTG to CII in a multi-ethnic population with HTGP. In our study, the time to reach the

**Table 4. Clinical and laboratory characteristics among the 54 patients with diabetes in the responder group who reached TG goal in less than 36 hours vs 36 hours or more.**

| | Less than 36 hours (n = 22) | 36 hours or More (n = 32) | P value |
|---|---|---|---|
| **Demographics:** | | | |
| Age (years) | 37.2 ± 7.7 | 39.3 ± 11.2 | 0.45 |
| Males, n (%) | 17 (77.3) | 20 (62.5) | 0.25 |
| Ethnicity, n (%) | | | 0.85 |
| Hispanic | 15 (68.2) | 20 (62.5) | |
| African American | 5 (22.7) | 8 (25.0) | |
| White | 2 (9.1) | 3 (9.4) | |
| Other | 0 (0.0) | 1 (3.1) | |
| BMI (kg/m$^2$) | 31.4 [27.0, 35.1] | 31.9 [28.3, 37.4] | 0.470 |
| Excess alcohol use, n (%) | 14 (63.6) | 11 (34.4) | 0.034* |
| **Baseline:** | | | |
| HbA1C% | 10.5 ± 2.6 | 11.2 ± 3.1 | 0.43 |
| Initial serum TG (mg/dL) | 2855.5 [1722.0, 4171.3] | 4814.5 [3403.8, 6990.0] | 0.001* |
| Serum TG at the start of CII (mg/dL) | 2493.0 [1527.5, 3902.3] | 4122.0 [3275.0, 6751.0] | 0.000* |
| Time from initial TG to start of CII (hours) | 3.0 [0.0. 5.3] | 2.5 [0.0, 5.0] | 0.813 |
| **Midpoint:** | | | |
| TG-50% (mg/dL) | 954.0 [585.5, 1305.8] | 1837.0 [1378.0, 2681.5] | 0.000* |
| Time from start of CII to TG-50% (hours) | 15.0 [6.0, 20.0] | 22.5 [13.3, 34.3] | 0.017* |
| Total insulin used to reach TG-50% (units) | 90.0 [45.8, 143.4] | 117.3 [81.6, 227.3] | 0.117 |
| **Final:** | | | |
| TG final (mg/dL) | 732.0 [546.8, 884.5] | 898.0 [674.0, 962.8] | 0.032* |
| Time from start of CII to TG goal (hours) | 23.5 [10.8, 28.5] | 54.0 [42.5, 72.5] | 0.000* |
| Total insulin used to reach TG goal (units) | 125.8 [64.3, 214.5] | 279.2 [234.0, 469.8] | 0.000* |
| Total daily insulin units used to reach TG goal (units/kg/day) | 1.9 [1.4, 2.3] | 1.6 [1.1, 1.8] | 0.019* |

*statistically significant P <0.05.

TG-goal was widely variable. The faster response of serum TG to CII was strongly associated with absence of DM, lower BMI, initial TG, and higher daily insulin use per kg body weight. In those without DM, the time to reach the TG-goal was about one day. Among those with DM, RRP had higher daily insulin use per kg body weight, lower initial TG and they reached the TG-goal within one day. In a literature review of 34 patients who received CII, patients with HTGP took three to five days to decrease serum TG to less than 500mg/dL [8]. In a study of 106 patients with HTGP receiving conservative management or CII, both groups reached TG <1000 mg/dL by day 3 [6]. This study did not find any statistical difference in TG response between the two groups. However, one of the notable limitations of that study is the use of sub-cutaneous insulin among the patients managed conservatively.

This study may suggest some practical management approaches to treating HTGP. In patients who have DM, we showed that much higher insulin/kg dose should be considered at initiation, at least 1.9 units/kg/day. As the management progresses, the time to reach TG-50% from initiation of CII can be utilized. At least half of the RRP reached TG-50% in 15 hours from initiation of CII compared to the much longer time taken by the SRP. Hence, an increase of insulin rate should be actively considered if, at approximately 15 hours after initiation of CII, TG did not decrease by 50%. In these circumstances, especially when some of the patients may not have DM, one should consider precautions for avoiding hypoglycemia. If TG is used

as basis for insulin adjustment, a parallel infusion of dextrose may be needed to maintain euglycemia with cautious monitoring.

The limitations of our study include that it was retrospective, was not randomized and without comparison to other treatments modalities. However, this was not the scope of this study. Additionally, the CII dose was adjusted based on blood glucose levels and not primarily determined by the TG response.

## Conclusion

CII was effective in reaching TG-goal in 95% of patients with HTGP. Half achieved TG-goal within 36 hours. Presence of DM, higher BMI and initial TG slowed the time to reach TG-goal. These baseline parameters and the rate of decline to TG-50% may be real-time indicators to initiate and adjust the CII for quicker response. Prospective studies are needed to establish more precise management protocols with emphasis on insulin dosing.

## Supporting information

**S1 Data.**
(XLSX)

## Author Contributions

**Conceptualization:** Vishnu Priya Pulipati, Ambika Amblee, Leon Fogelfeld.

**Data curation:** Vishnu Priya Pulipati, Ambika Amblee, Sara Elizabeth T. Yap, Bettina Tahsin, Leon Fogelfeld.

**Formal analysis:** Vishnu Priya Pulipati, Ambika Amblee, Hafeez Shaka, Bettina Tahsin, Leon Fogelfeld.

**Investigation:** Vishnu Priya Pulipati, Ambika Amblee, Sara Elizabeth T. Yap, Hafeez Shaka, Leon Fogelfeld.

**Methodology:** Vishnu Priya Pulipati, Ambika Amblee, Sara Elizabeth T. Yap, Hafeez Shaka, Bettina Tahsin, Leon Fogelfeld.

**Project administration:** Ambika Amblee, Hafeez Shaka, Leon Fogelfeld.

**Supervision:** Ambika Amblee, Leon Fogelfeld.

**Validation:** Ambika Amblee, Hafeez Shaka, Bettina Tahsin, Leon Fogelfeld.

**Visualization:** Ambika Amblee, Bettina Tahsin, Leon Fogelfeld.

**Writing – original draft:** Vishnu Priya Pulipati, Ambika Amblee, Bettina Tahsin, Leon Fogelfeld.

**Writing – review & editing:** Vishnu Priya Pulipati, Ambika Amblee, Bettina Tahsin, Leon Fogelfeld.

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
