## [Decision Letter · Decision Letter 0]

14 Sep 2021

PONE-D-21-25261Hypertriglyceridemia-Associated Acute Pancreatitis: Response to Continuous Insulin InfusionPLOS ONE

Dear Dr. Amblee,

Thank you for submitting your manuscript to PLOS ONE. After careful consideration, we feel that it has merit but does not fully meet PLOS ONE’s publication criteria as it currently stands. Therefore, we invite you to submit a revised version of the manuscript that addresses the points raised during the review process.

Your paper has been reviewed by three experts in the field. Notably they have mentioned that the methods section should be expanded by providing further details (e.g. concerning i.v. hydration or continuous insulin infusion). Please also check your manuscript for typos.

We look forward to receiving your revised manuscript.

Kind regards,

Zoltán Rakonczay Jr., M.D., Ph.D., D.Sc.

Academic Editor

PLOS ONE

Journal Requirements:

Reviewers' comments:

Reviewer's Responses to Questions

**Comments to the Author**

1. Is the manuscript technically sound, and do the data support the conclusions?

Reviewer #1: Yes

Reviewer #2: Yes

Reviewer #3: Yes

2. Has the statistical analysis been performed appropriately and rigorously? 

Reviewer #1: Yes

Reviewer #2: Yes

Reviewer #3: Yes

3. Have the authors made all data underlying the findings in their manuscript fully available?

Reviewer #1: No

Reviewer #2: Yes

Reviewer #3: Yes

4. Is the manuscript presented in an intelligible fashion and written in standard English?

Reviewer #1: Yes

Reviewer #2: Yes

Reviewer #3: Yes

5. Review Comments to the Author

Reviewer #1: The aim of this retrospective case series is to analyze the outcome of to continuous insulin infusion in hypertriglyceridemia-associated acute pancreatitis. It was concluded that continuous insulin infusion was effective in reducing serum triglyceride level in the majority of patients. Presence of diabetes, higher BMI and initial triglyceride slowed the time needed to effectively reduce serum triglyceride level.

There are many case reports and series demonstrating the effectiveness of insulin in lowering triglyceride level. This study provides data about the quantity of insulin and the duration required to reduce effectively the serum triglyceride level.

Criticisms:

1. Fasting and intravenous hydration applied in the treatment of acute pancreatitis rapidly and greatly reduces serum triglyceride level and can be as effective as intravenous insulin. Provide data about the applied intravenous fluid therapy in the first days of your patients. Intravenous hydration can be a confounding factor in this study. Please, discuss it.

2. Was dextrose infusion applied as part of the parenteral fluid therapy in the study? Please, provide details in the Methods.

3. Could you give data about the severity and outcome of acute pancreatitis especially in the RP and NR, SRP and RRP group, respectively?

4. What is the 75th percentile in this data from table 2? 4 [0.5]

5. Page 12: ‘Figure 1: Consort Table’. Some contradiction: is it a figure or table? ‘Patient’s flow chart’ is suggested to be used.

6. Page 22: ‘Figure 1’ is supposed to be Figure 2. What is the meaning of gosl? Please use the word time/Time, insulin/Insulin consistently.

7. The style of references does not follow the requirements of the journal.

8. There are many typos in the manuscript. Some of them:

Abstract: kg/m2…instead of …kg/m 2…

Abstract: …HbA1c… instead of …HbA1C…

Page 10: …abdominal pain… instead of…. pain abdomen…

Repeatedly:…. p = 0.130… instead of …p 0.130…

Table 2 title: …. less than 36 hours vs 36 hours …instead of … less than vs 36 hours …

Table 2:… 36 hours or more… instead of … 36 hours or More …

Table 2: … (hours) … instead of … hours)…

Reviewer #2: 1. Please review abstract. Currently not able to be interpreted easily.

2. "CII is an effective modality of therapy to reduce acutely the TG levels" - please review syntax

3. "AP was defined per Atlanta classification of acute

pancreatitis 2012 as any two of the three following criteria: pain abdomen (acute onset, persistent, severe, epigastric pain often radiating to the back), pancreatic enzymes (serum lipase or amylase) level at least 3 times greater than the upper limit of normal, computed tomography (CT) or magnetic resonance imaging (MRI) evidence of acute pancreatitis" - please correct to abdominal pain

4. Why were pregnant women excluded from this study?

5. "Secondary outcomes were time durations to achieve TG-50% (half of the respective initial TG) and time durations to achieve TG-goal" - please take out durations

6. What is your institution's protocol about rate and concentration of dextrose delivery? This will affect insulin doses

7. "The four patients who did not reach TG-goal (NR, non-responders) had a long time to initiation of CII (3 [0, 5.5] vs 10 [2.2, 37.2] hr, p 0.130) but otherwise did not differ from RP" - p value suggests this is not significant

Reviewer #3: In this manuscript authors investigated efficacy of continuous insulin insfusion on lowering triglyceride levels in patients with hypertriglyceridemic pancreatitis. They also search for the factors that can predict treatment success. This study gives a new insight how insulin, as a triglyceride lowering agent, could be used in patients with hypertriglyceridemic pancreatitis and which parameters could help in titration of therapy. In the current literature this kind of data are lacking so this manuscript is of great importance from the clinical and research points of view.

Introduction provides good and concise background of the topic. The objective is clearly defined.

Methodology of the study is well designed but continuous insulin infusion protocol could be explained in a more detailed way (quantity of insulin in infusion, type and volume of solution, frequency of glucose checks, goal glucose levels and adjustment of infusion rate). Other components of methodology, inclusion/exclusion criteria and outcomes are thoroughly written.

Results are well presented. Tables and figures are clear, informative and easy to follow. The data provide enough evidence to make a conclusions.

Discussion gives good perspective of current knowledge and how findings of this study can upgrade clinical practice. Conclusions are well presented and supported by the data.

The literature cited is relevant to the study.

6. PLOS authors have the option to publish the peer review history of their article (what does this mean?). If published, this will include your full peer review and any attached files.

Reviewer #1: No

Reviewer #2: No

Reviewer #3: No

---

## [Author Response · Author response to Decision Letter 0]

22 Oct 2021

Journal Requirements:

Data included as Supplemental Materials

Information about de-identified data added to Methods section, page 4.

Reviewers' comments:

Reviewer's Responses to Questions

Comments to the Author

1. Is the manuscript technically sound, and do the data support the conclusions?

Reviewer #1: Yes

Reviewer #2: Yes

Reviewer #3: Yes

2. Has the statistical analysis been performed appropriately and rigorously?

Reviewer #1: Yes

Reviewer #2: Yes

Reviewer #3: Yes

3. Have the authors made all data underlying the findings in their manuscript fully available?

Reviewer #1: No

Reviewer #2: Yes

Reviewer #3: Yes

4. Is the manuscript presented in an intelligible fashion and written in standard English?

Reviewer #1: Yes

Reviewer #2: Yes

Reviewer #3: Yes

5. Review Comments to the Author

Reviewer #1: The aim of this retrospective case series is to analyze the outcome of to continuous insulin infusion in hypertriglyceridemia-associated acute pancreatitis. It was concluded that continuous insulin infusion was effective in reducing serum triglyceride level in the majority of patients. Presence of diabetes, higher BMI and initial triglyceride slowed the time needed to effectively reduce serum triglyceride level.

There are many case reports and series demonstrating the effectiveness of insulin in lowering triglyceride level. This study provides data about the quantity of insulin and the duration required to reduce effectively the serum triglyceride level.

Criticisms:

1. Fasting and intravenous hydration applied in the treatment of acute pancreatitis rapidly and greatly reduces serum triglyceride level and can be as effective as intravenous insulin. Provide data about the applied intravenous fluid therapy in the first days of your patients. Intravenous hydration can be a confounding factor in this study. Please, discuss it.

In our ICUs, such patients would receive IV fluids (either 0.9% or 0.45% saline) based on hydration parameters. We did not include these data since the IV fluid therapy is consistently implemented with hydration information added in the Methods section, page 4.

 

2. Was dextrose infusion applied as part of the parenteral fluid therapy in the study? Please, provide details in the Methods.

There were no indications in the blood glucose data that any patients would have required dextrose infusions, therefore not mentioned in the Method section.

3. Could you give data about the severity and outcome of acute pancreatitis especially in the RP and NR, SRP and RRP group, respectively?

This study was focused on the impact of CII on HTG reduction. Data to assess severity of AP were not collected. However, based on admission and discharge dates, all patients survived and were discharged home. Sentence added in Results, page 10.

4. What is the 75th percentile in this data from table 2? 4 [0.5]

Fixed typo – 4 [0, 5]

5. Page 12: ‘Figure 1: Consort Table’. Some contradiction: is it a figure or table? ‘Patient’s flow chart’ is suggested to be used.

Figure 1 title changed to Consort Chart

6. Page 22: ‘Figure 1’ is supposed to be Figure 2. What is the meaning of gosl? Please use the word time/Time, insulin/Insulin consistently.

Figure 1 changed to Figure 2 and typos/inconsistencies fixed.

7. The style of references does not follow the requirements of the journal.

Reference style updated.

8. There are many typos in the manuscript. Some of them:

Abstract: kg/m2…instead of …kg/m 2… 

Abstract: …HbA1c… instead of …HbA1C…

Page 10: …abdominal pain… instead of…. pain abdomen… 

Repeatedly:…. p = 0.130… instead of …p 0.130… 

Table 2 title: …. less than 36 hours vs 36 hours …instead of … less than vs 36 hours …

Table 2:… 36 hours or more… instead of … 36 hours or More …

Table 2: … (hours) … instead of … hours)…

Typos corrected.

Reviewer #2: 1. Please review abstract. Currently not able to be interpreted easily.

Abstract revised.

2. "CII is an effective modality of therapy to reduce acutely the TG levels" - please review syntax

Sentence on page 3 revised to: CII is an effective modality of therapy to reduce TG levels rapidly. 

3. "AP was defined per Atlanta classification of acute

pancreatitis 2012 as any two of the three following criteria: pain abdomen (acute onset, persistent, severe, epigastric pain often radiating to the back), pancreatic enzymes (serum lipase or amylase) level at least 3 times greater than the upper limit of normal, computed tomography (CT) or magnetic resonance imaging (MRI) evidence of acute pancreatitis" - please correct to abdominal pain

Corrected.

4. Why were pregnant women excluded from this study?

Pregnant women were excluded to eliminate pregnancy as a confounding variable. At the same time, no patients with pregnancy were in the initial study group prior to exclusion. In this study of acute pancreatitis, no pregnant patients with acute pancreatitis presented.

5. "Secondary outcomes were time durations to achieve TG-50% (half of the respective initial TG) and time durations to achieve TG-goal" - please take out durations

Corrected.

6. What is your institution's protocol about rate and concentration of dextrose delivery? This will affect insulin doses

No information about rate and concentration of dextrose delivery is needed so there were no indications in the blood glucose data that any patients would have required dextrose infusions.

 

7. "The four patients who did not reach TG-goal (NR, non-responders) had a long time to initiation of CII (3 [0, 5.5] vs 10 [2.2, 37.2] hr, p 0.130) but otherwise did not differ from RP" - p value suggests this is not significant

As suggested, this statement was deleted.

Reviewer #3: In this manuscript authors investigated efficacy of continuous insulin insfusion on lowering triglyceride levels in patients with hypertriglyceridemic pancreatitis. They also search for the factors that can predict treatment success. This study gives a new insight how insulin, as a triglyceride lowering agent, could be used in patients with hypertriglyceridemic pancreatitis and which parameters could help in titration of therapy. In the current literature this kind of data are lacking so this manuscript is of great importance from the clinical and research points of view.

Introduction provides good and concise background of the topic. The objective is clearly defined.

Methodology of the study is well designed but continuous insulin infusion protocol could be explained in a more detailed way (quantity of insulin in infusion, type and volume of solution, frequency of glucose checks, goal glucose levels and adjustment of infusion rate). 

More details were added to the Methods section, page 4.

Other components of methodology, inclusion/exclusion criteria and outcomes are thoroughly written.

Results are well presented. Tables and figures are clear, informative and easy to follow. The data provide enough evidence to make a conclusions.

Discussion gives good perspective of current knowledge and how findings of this study can upgrade clinical practice. Conclusions are well presented and supported by the data.

The literature cited is relevant to the study.

6. PLOS authors have the option to publish the peer review history of their article (what does this mean?). If published, this will include your full peer review and any attached files.

Do you want your identity to be public for this peer review? For information about this choice, including consent withdrawal, please see our Privacy Policy.

Reviewer #1: No

Reviewer #2: No

Reviewer #3: No

---

## [Decision Letter · Decision Letter 1]

11 Nov 2021

Hypertriglyceridemia-Associated Acute Pancreatitis: Response to Continuous Insulin Infusion

PONE-D-21-25261R1

Dear Dr. Amblee,

We’re pleased to inform you that your manuscript has been judged scientifically suitable for publication and will be formally accepted for publication once it meets all outstanding technical requirements.

Kind regards,

Zoltán Rakonczay Jr., M.D., Ph.D., D.Sc.

Academic Editor

PLOS ONE

Additional Editor Comments (optional):

Reviewers' comments:

Reviewer's Responses to Questions

**Comments to the Author**

1. If the authors have adequately addressed your comments raised in a previous round of review and you feel that this manuscript is now acceptable for publication, you may indicate that here to bypass the “Comments to the Author” section, enter your conflict of interest statement in the “Confidential to Editor” section, and submit your "Accept" recommendation.

Reviewer #1: All comments have been addressed

Reviewer #3: All comments have been addressed

2. Is the manuscript technically sound, and do the data support the conclusions?

Reviewer #1: Yes

Reviewer #3: (No Response)

3. Has the statistical analysis been performed appropriately and rigorously? 

Reviewer #1: I Don't Know

Reviewer #3: (No Response)

4. Have the authors made all data underlying the findings in their manuscript fully available?

Reviewer #1: Yes

Reviewer #3: (No Response)

5. Is the manuscript presented in an intelligible fashion and written in standard English?

Reviewer #1: Yes

Reviewer #3: (No Response)

6. Review Comments to the Author

Reviewer #1: All required questions have been answered.

All required questions have been answered.

All required questions have been answered.

All required questions have been answered.

Reviewer #3: (No Response)

7. PLOS authors have the option to publish the peer review history of their article (what does this mean?). If published, this will include your full peer review and any attached files.

Reviewer #1: No

Reviewer #3: No

---

## [Editor Report · Acceptance letter]

18 Nov 2021

PONE-D-21-25261R1 

Hypertriglyceridemia-Associated Acute Pancreatitis: Response to Continuous Insulin Infusion 

Dear Dr. Amblee:

I'm pleased to inform you that your manuscript has been deemed suitable for publication in PLOS ONE. Congratulations! Your manuscript is now with our production department. 

Kind regards, 

on behalf of

Dr. Zoltán Rakonczay Jr. 

Academic Editor

PLOS ONE